

# Segal's Law, 16S rRNA gene sequencing, and the perils of foodborne pathogen detection within the American Gut Project

James B. Pettengill and  Hugh Rand

Biostatistics and Bioinformatics Staff, Office of Analytics and Outreach, US Food and Drug Administration, College Park, MD, United States of America

## ABSTRACT

Obtaining human population level estimates of the prevalence of foodborne pathogens is critical for understanding outbreaks and ameliorating such threats to public health. Estimates are difficult to obtain due to logistic and financial constraints, but citizen science initiatives like that of the American Gut Project (AGP) represent a potential source of information concerning enteric pathogens. With an emphasis on genera *Listeria* and *Salmonella*, we sought to document the prevalence of those two taxa within the AGP samples. The results provided by AGP suggest a surprising 14% and 2% of samples contained *Salmonella* and *Listeria*, respectively. However, a reanalysis of those AGP sequences described here indicated that results depend greatly on the algorithm for assigning taxonomy and differences persisted across both a range of parameter settings and different reference databases (i.e., Greengenes and HITdb). These results are perhaps to be expected given that AGP sequenced the V4 region of 16S rRNA gene, which may not provide good resolution at the lower taxonomic levels (e.g., species), but it was surprising how often methods differ in classifying reads—even at higher taxonomic ranks (e.g., family). This highlights the misleading conclusions that can be reached when relying on a single method that is not a gold standard; this is the essence of Segal's Law: an individual with one watch knows what time it is but an individual with two is never sure. Our results point to the need for an appropriate molecular marker for the taxonomic resolution of interest, and calls for the development of more conservative classification methods that are fit for purpose. Thus, with 16S rRNA gene datasets, one must be cautious regarding the detection of taxonomic groups of public health interest (e.g., culture independent identification of foodborne pathogens or taxa associated with a given phenotype).

Corresponding author
James B. Pettengill,
fixtgear@gmail.com,
james.pettengill@fda.hhs.gov

## INTRODUCTION

Obtaining estimates of the prevalence of foodborne pathogens is critical to public health because the estimates affect policy, allocation of funding, and understanding of foodborne outbreaks. Such figures are difficult to obtain but estimates do exist based on surveillance

programs. Estimates by the US Center for Disease Control of the number of cases in 2013 of *Salmonella* and *Listeria* was 7,277 (incidence $= 15.19$) and 123 (incidence $= 0.26$), respectively, per 100,000 individuals in the United States (http://www.cdc.gov/mmwr/preview/mmwrhtml/mm6315a3.htm). These numbers are based on the active monitoring done through FoodNet (Foodborne Diseases Active Surveillance Network) and come from laboratory-confirmed infections in 10 US localities that account for about 15% of the US population. Such monitoring has provided extremely valuable data to help evaluate prevention measures, understand the causes and transmission of food-borne diseases, and set targets for improving the domestic food supply (*Centers for Disease and Prevention, 2000*; *Watstein & Jovanovic, 2003*). However, monitoring of the type done by FoodNet is not without caveats, chief among them are the underreporting of cases due to infected individuals not visiting local health facilities when sickened with a foodborne pathogen.

A complementary approach to large-scale surveillance networks managed by state and federal agencies is the solicitation of the general population to participate in a study or data collection initiative; such information may provide an independent estimate of the prevalence of enteric pathogens such as *Listeria* and *Salmonella*. One such initiative is the American Gut Project (AGP; http://www.americangut.org), which describes itself as "…one of the world's largest crowd sourced, citizen science projects in the country (USA). We discover new information daily to shed light on the connection between the human microbiome and health." The AGP has performed sequencing of the V4 region of the 16S rRNA gene from thousands of individual participant's skin, mouth, and/or feces. Such an approach is not without caveats, however. Chief among them are the potential bias introduced by the non-random participation by citizens (i.e., convenience sampling), the computational and bioinformatic challenges associated with analyzing the large amount of DNA sequence data, and the potential for misinterpretation (*Gonzalez et al., 2016*). Nevertheless, the data that is produced may be quite informative about public health issues related to foodborne pathogens by detecting carriers and asymptomatic cases that hospital-centric sampling schemes would miss.

Here we describe our efforts to use the data produced by the AGP to determine the prevalence of *Listeria* and *Salmonella*—two genera with species of public health interest as causative agents of gastroenteritidis (or worse such as death and spontaneous abortion in women). Given the rich metadata associated with the human subjects that participated in the AGP, we were particularly interested in determining life style habits or other factors that correlate strongly with individuals that are positive for particular pathogens; we were encouraged by the publicly available results provided by the AGP that showed 14% and 2% of samples did contain either *Salmonella* or *Listeria*, respectively (ftp://ftp.microbio.me/AmericanGut//ag-July-29-2016//07-taxa/notrim/otu_table_fecal_L6.biom). However, in attempting to validate those results we found that taxonomic classification of reads varied greatly depending on the algorithm and reference database used. Our objective here is to describe those additional analyses and discuss the implications of our results for using such data for public health surveillance.

**Table 1  Summary of the four different methods used to assign taxonomy to reads.** Default parameter settings are provided and "additional" values represent those used to determine the influence of settings on the congruence between methods.

| Classifier | Description | Parameter settings |
|---|---|---|
| BLAST | Assignment is based on the best hit from BLAST (*Altschul et al., 1990*) | blast_e_value<br>  default: 0.001<br>  additional: $1e-20$, $1e-50$, $1e-65$ |
| RDP | Assignment via a Naïve Bayes Classifier under which reads are parsed into 8-mers (*Wang et al., 2007*) | confidence<br>  default: 0.5<br>  additional: *0.6, 0.7, 0.8* |
| SortMeRNA | Assignment is based on a variant of the Longest Increasing Subsequence for string matching within which alignment quality is based on *E*-values (*Kopylova et al., 2014*) | min_consensus_fraction<br>  default: 0.51<br>  additional: *0.75, 0.9*<br>similarity: 0.9<br>sortmerna_e_value: 1.0<br>sortmerna_coverage: 0.9<br>sortmerna_best_N_alignments: 5 |
| UCLUST | UCLUST clustering algorithm in conjunction with a reference database (*Edgar, 2010*) | min_consensus_fraction<br>  default: 0.51<br>  additional: *0.68*<br>similarity: 0.9<br>uclust_max_accepts: 3 |

# MATERIALS AND METHODS

## American Gut Project data

American Gut Project data as of August 1, 2016 were analyzed. Given that our original objective was to determine the percentage of samples that contained either *Salmonella* or *Listeria* we queried the QIIME output files (i.e., ftp://ftp.microbio.me/AmericanGut//ag-July-29-2016/07-taxa/notrim/ag-cleaned_L6.txt) provided by AGP for those taxa. The AGP data contained 10,294 fecal, skin, or oral samples, and yielded 188 samples positive for *Listeria* and 1,495 positive for *Salmonella*; 31 samples were positive for both. This yielded 1,652 samples that were carried forward and analyzed here to determine the robustness of these classifications across multiple taxonomy assigners. The sequences from each of the 1,652 were extracted from the fasta file available from AGP (ftp://ftp.microbio.me/AmericanGut//ag-July-29-2016/02-filtered/sequences-notrim.fna), which resulted in a 48,312,131 sequences to be classified. No additional filtering of reads was done.

## Taxonomic assigners

For each of the 1,652 samples, we classified the reads associated with them to taxonomy using four different classifiers—RDP (*Wang et al., 2007*), UCLUST (*Edgar, 2010*), BLAST (*Altschul et al., 1990*), and SortMeRNA (*Kopylova et al., 2014*) (Table 1). SortMeRNA was the method used to produce the AGP results. The program QIIME v1.9.1 (*Caporaso et al., 2010*) was used to run each of these classifiers. These assigners do not represent an exhaustive list of those available and were chosen as they are among the most frequently used; additional classifiers are likely to only increase the variability that we observe and,

therefore, not change our findings. Additionally, we make no statements about the accuracy of the methods as we do not know the true taxonomy of the reads but rather are estimating the variability in classification across the methods, which illustrates the degree to which conclusions about pathogen presence is dependent on the classifier used.

Within each classifier a range of cutoffs were used for the parameter that measures confidence of taxonomic assignment (Table 1). The cutoff settings for these parameters are specific to each assigner and were determined through exploratory analyses that evaluated which cutoff values result in a substantial difference in number of reads classified. An example is UCLUST, which has a min_consensus_fraction parameter, which determines the fraction of hits that must agree in the number of assignments considered (max_accepts). With max_accepts set to 3 (the default), all that matters is whether min_consensus_fraction is more or less than 2/3. Thus, we evaluated two settings (1) min_consensus_fraction = 0.51 (the default, at least 2/3 agreement is required) and (2) min_consensus_fraction = 0.68 (complete agreement is required). The latter threshold is more conservative (higher specificity) but also runs the risk of a higher false-negative rate (lower sensitivity).

## Reference databases

Analyses were performed against two different reference databases. The first database was the Greengenes (gg_13_8) database (*DeSantis et al., 2006*). This database was used to produce the AGP results and we primarily focus on the results using this database. Within the Greengenes database there are 99,322 sequences of which four are classified to *Salmonella* (one to only genus and three to species *enterica*) and five sequences classified to *Listeria* (one to each of the following species *monocytogenes*, *seeligeri*, *grayi*, *fleischmannii*, *weihenstephanensis*).

The second database was the human microbiome specific reference database HITdb described in *Ritari et al. (2015)*. HITdb includes many fewer sequences than Greengenes ($n = 2,472$) of which a single sequence is classified to *Salmonella enterica* and zero are classified to the genus *Listeria*. The fewer number of representatives at this taxonomic level for these pathogens, and likely others, may be a desirable property as it could avoid false-positives. However, such a smaller database relative to Greengenes may run the risk of being too restrictive where a read could be reliably classified at one of those taxa given a sufficient algorithm. The taxonomy of within HITdb begins at the phylum level and, therefore, we do not present congruence among assigners at that kingdom level when using HITdb.

## Congruence between classifiers

The results from each of the classifiers with the various parameter settings were combined in a pairwise fashion. To evaluate the congruence between the classifiers, we calculated the percent of total reads within each of five different comparison categories (1) match: both classifiers assigned the read to the same taxon, (2) mismatch: the classifiers differed in the taxonomy assigned to a read, (3 & 4) method 1 unclassified and method 2 unclassified: cases in which one classifier did not assign the read to a taxon but the other method did, and (5) noMatch: not classified by either classifier. The small fraction of the reads in the noMatch category (~0.8% for the default parameter settings using Greengenes) were not

analyzed. As these categories suggest, we include reads classified to the species level and reads partially classified (e.g., the lowest taxonomic rank assigned to a read was only phylum or family). We also explored more explicit methods for determining congruence between classifiers (i.e., inter-rater agreement; e.g., Cohen's κ (*Cohen, 1960*)). Such statistics attempt to account for the fact that two classifiers could agree by chance, which percent of reads that match (like that used here) does not. However, preliminary analyses showed these statistics to be difficult to interpret as a result of the large number of observations and at times uneven distribution of classifications across a small number of categories (e.g., at the kingdom or phylum level). For example, at higher taxonomic ranks there is clearly better agreement between the classifiers (see Results) but prevalence and bias adjusted Cohen's κ (PABAK) was substantially less than that observed at the species level (Supplemental Information 1).

As an example of how the incongruence among classifiers impacts the detection of a specific taxon, we focused on the number of reads under each method that were classified to the genera *Salmonella* and *Listeria* and quantified differences between classifiers. This was done for only the results produced with Greengenes. The congruence between methods in which reads were classified to *Salmonella* and *Listeria* was evaluated using the Jaccard Similarity index (0 meaning two methods classified completely different reads to a given taxon; 1 denotes complete overlap in the reads classified to a given taxon).

## RESULTS

### Default settings and Greengenes database

The initial investigation into taxonomy assignments provided by the AGP indicated 188 samples positive for *Listeria,* 1,495 samples positive for *Salmonella*, and 31 samples positive for both, yielding 1,652 samples. Taxonomic resolution was only to genus so the AGP results do not include whether pathogenic *Listeria* (e.g., *Listeria monocytogenes*) was present. To determine the robustness of these classifications we analyzed the sequences from those samples using default parameter settings and the Greengenes database (similar to the AGP analyses) and found little congruence among the four classifiers investigated in the assignment of reads to lower taxonomic ranks. For example, in all but the highest taxonomic rank (kingdom), there was an average difference of approximately 30% in the number of unique taxa detected (Fig. 1). The pattern of difference in number of taxa detected was also not consistent across the ranks where, for instance, the RDP classifier detected more taxa than SortMeRNA at the genus and species level but not at the phylum, class, order or family level (Fig. 1).

Focusing on congruence in the taxonomic identity assigned to reads, we found that congruence among the classifiers decreased with decreasing taxonomic rank (Fig. 2). This suggests that it is more difficult and there is greater error associated with classifying reads to lower taxonomic ranks, which has also been observed in other studies (*Mizrahi-Man, Davenport & Gilad, 2013*; *Wang et al., 2007*). This incongruence was primarily driven by a read being classified to a taxonomic group with one method but not the other rather than each method classifying the same read to different taxonomic groups (Fig. 2). Certain

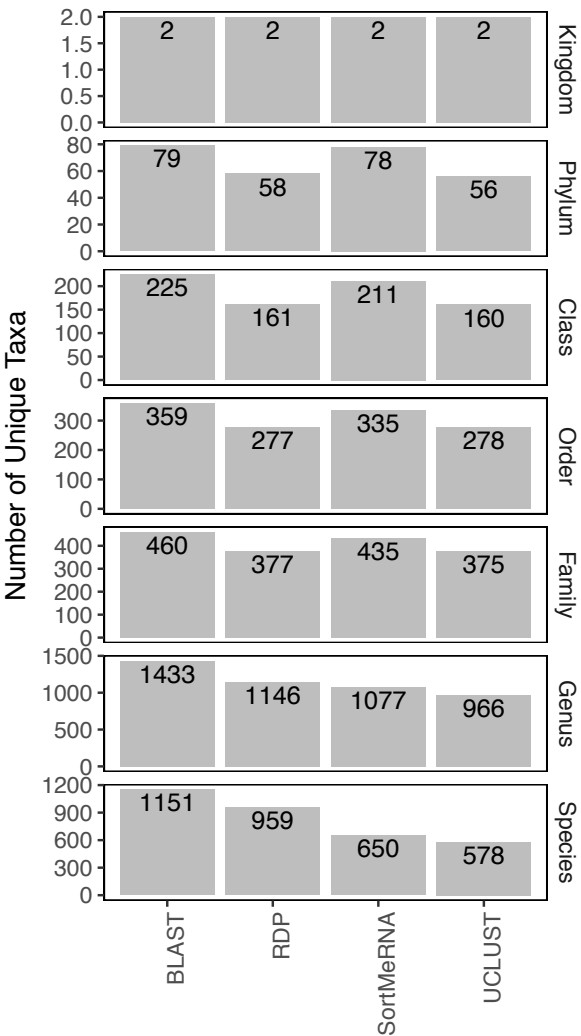

**Figure 1  The number of unique taxa to which reads were assigned by each classification method.** The results are grouped by taxonomic rank and represent those obtained using default parameter settings and the Greengenes database.

methods also seemed to result in greater incongruence. For example, RDP classified many more reads at the genus and species levels than the other methods (Fig. 2).

When we focused on the congruence across the classifiers in assigning reads to *Salmonella* or *Listeria*, we found that 447 and 1,521 samples contained reads classified to *Listeria* or *Salmonella*, respectively. The number of *Listeria* 'positive' samples reported by AGP was 188 and for *Salmonella* it was 1,495 (Table 2). However, the different methods varied substantially in the number of reads classified to each genus with BLAST consistently being the most liberal (i.e., classified the most reads) followed by UCLUST; RDP was the most conservative (i.e., did not classify reads) (Table 2). Perhaps of more importance is that there was little congruence between the methods in which reads were classified to each taxonomic group as indicated by the low Jaccard similarity indices between the pairwise comparisons of the methods (Fig. 3). The metric suggests there is little consistency among the methods
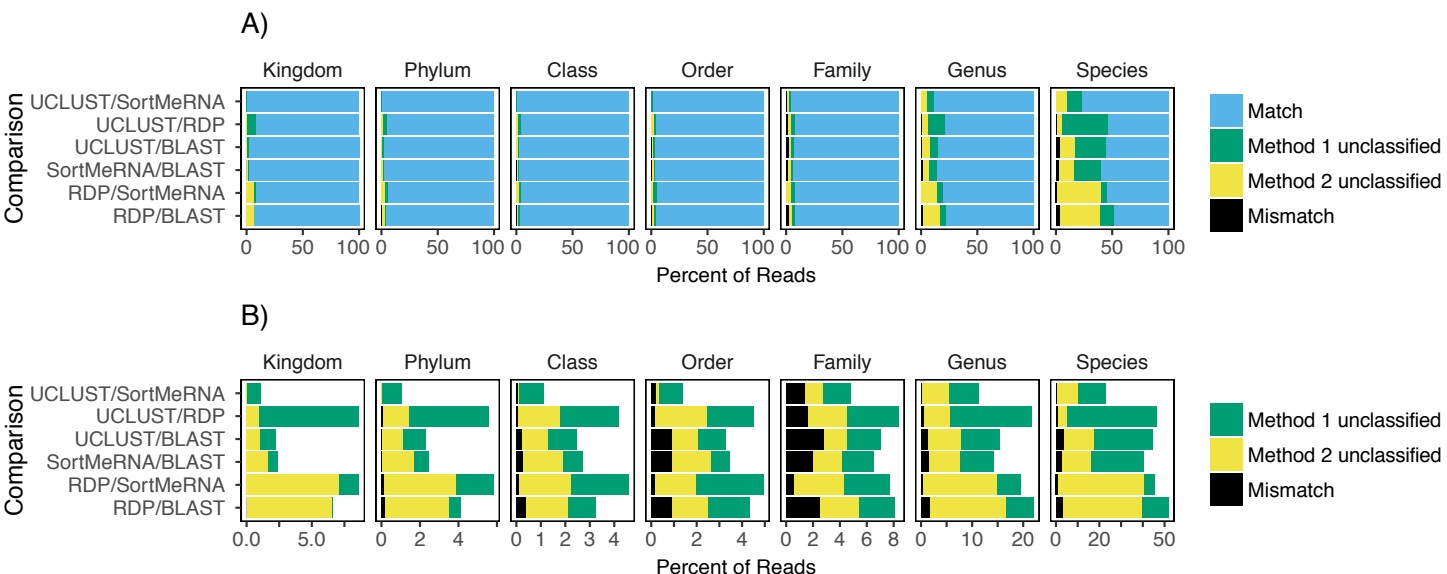

**Figure 2** **The agreement between taxon assignments, grouped by taxonomic rank.** Shown are the percentage of reads within each pairwise comparison among the classification methods that either matched, mismatched, or was not classified by one of the two methods at the default settings and using the Greengenes reference database. For the unclassified designation method 1 refers to the first method listed in the comparison; the second method is the second one listed (e.g., in the RDP/BLAST comparison method 1 is RDP and method 2 is BLAST). Results are shown for all congruence categories (A) and, in order to better see the differences among the mismatch and unclassified categories, with match removed (B).

when classifying *Listeria* or *Salmonella* (genus or species) to the reads. For example, only 22% of the reads classified by UCLUST or BLAST to a species within *Salmonella* were actually classified to the same species—some of those reads may not have been classified at all by one method while others could have been classified to a different species (Fig. 3).

Some of these differences in classifications could likely be reconciled by changing the confidence parameter. For example, with the default settings, read 10317.000001040_2356 was clustered by UCLUST and BLAST to the genus *Salmonella,* by RDP to the species *Salmonella enterica,* and by SortMeRNA to the family Enterobacteriaceae. If one reduced the confidence setting for UCLUST, BLAST and SortMeRNA then the methods might converge on the classification of *Salmonella enterica.* Alternatively, the confidence settings for UCLUST, BLAST, and RDP could be increased so the methods would converge on the same classification at the family level. However, looking at read 10317.000001127_16840, it was classified to the genus *Enterococcus* within the order Lactobacillales by SortMeRNA and UCLUST, only to the class level (Bacilli) with RDP (even with the low default confidence threshold of 0.5), and *Listeria weihenstephanensis* within the order Bacillales by BLAST. This read, which is representative of many of the reads classified to *Listeria* by at least one method, cannot be congruently classified below the rank of class. Although these examples illustrate the ways in which differences in classification may be reconciled, this would be problematic for an entire set of reads (e.g., each read may require its own threshold settings and reads may be 'under' classified if one takes a lowest common ancestor approach).

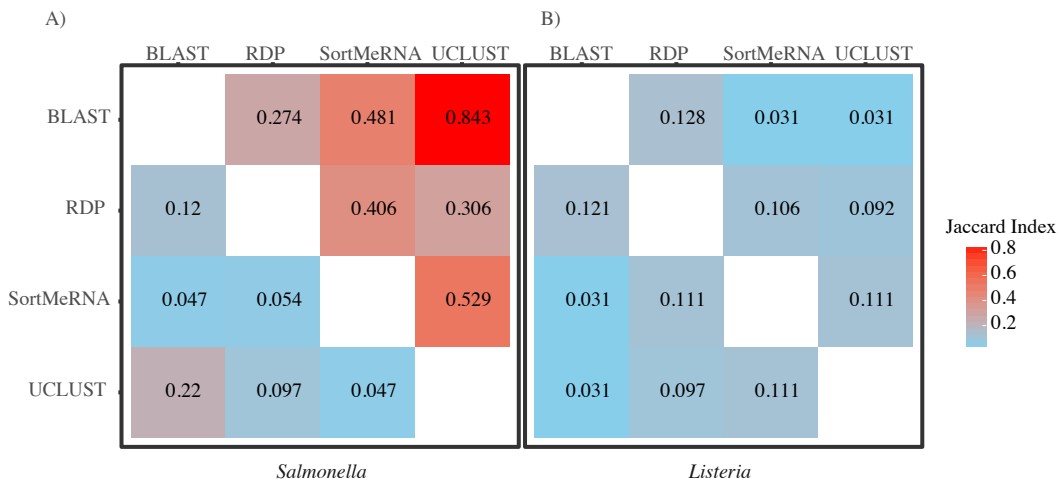

**Figure 3** **A heatmap of Jaccard Indices showing the low degree of congruence between the methods in which reads were classified to (A)** *Salmonella* **and (B)** *Listeria*. Values in the upper triangle are indices for comparison at the genus level; values in the lower triangle are for comparisons at the species level. A Jaccard Index of 1 indicates complete agreement in the reads classified to a given taxon for two methods; a Jaccard index of 0 indicates no overlap in the reads classified to a given taxon for two methods.

**Table 2** **The number of reads assigned to species within the genera** *Salmonella* **and** *Listeria* **and the number of samples 'positive' for each.** Results are those obtained using default parameter settings and the Greengenes database.

| Taxon | Classifier | Number of reads classified | Number of 'positive' samples |
|---|---|---|---|
| *Listeria fleischmannii* [a] | BLAST | 7 | 7 |
| | RDP | 0 | 0 |
| | SORTMERNA | 0 | 0 |
| | UCLUST | 0 | 0 |
| *Listeria grayi* [a] | BLAST | 85 | 39 |
| | RDP | 0 | 0 |
| | SORTMERNA | 4 | 4 |
| | UCLUST | 6 | 6 |
| *Listeria monocytogenes* | BLAST | 1 | 1 |
| | RDP | 0 | 0 |
| | SORTMERNA | 0 | 0 |
| | UCLUST | 0 | 0 |
| *Listeria weihenstephanensis* [a] | BLAST | 3,753 | 426 |
| | RDP | 80 | 54 |
| | SORTMERNA | 14 | 12 |
| | UCLUST | 9 | 8 |
| *Salmonella enterica* | BLAST | 457 | 204 |
| | RDP | 72 | 57 |
| | SORTMERNA | 43 | 40 |
| | UCLUST | 2,090 | 522 |

**Notes.**
[a] Non-pathogenic and non-hemolytic.

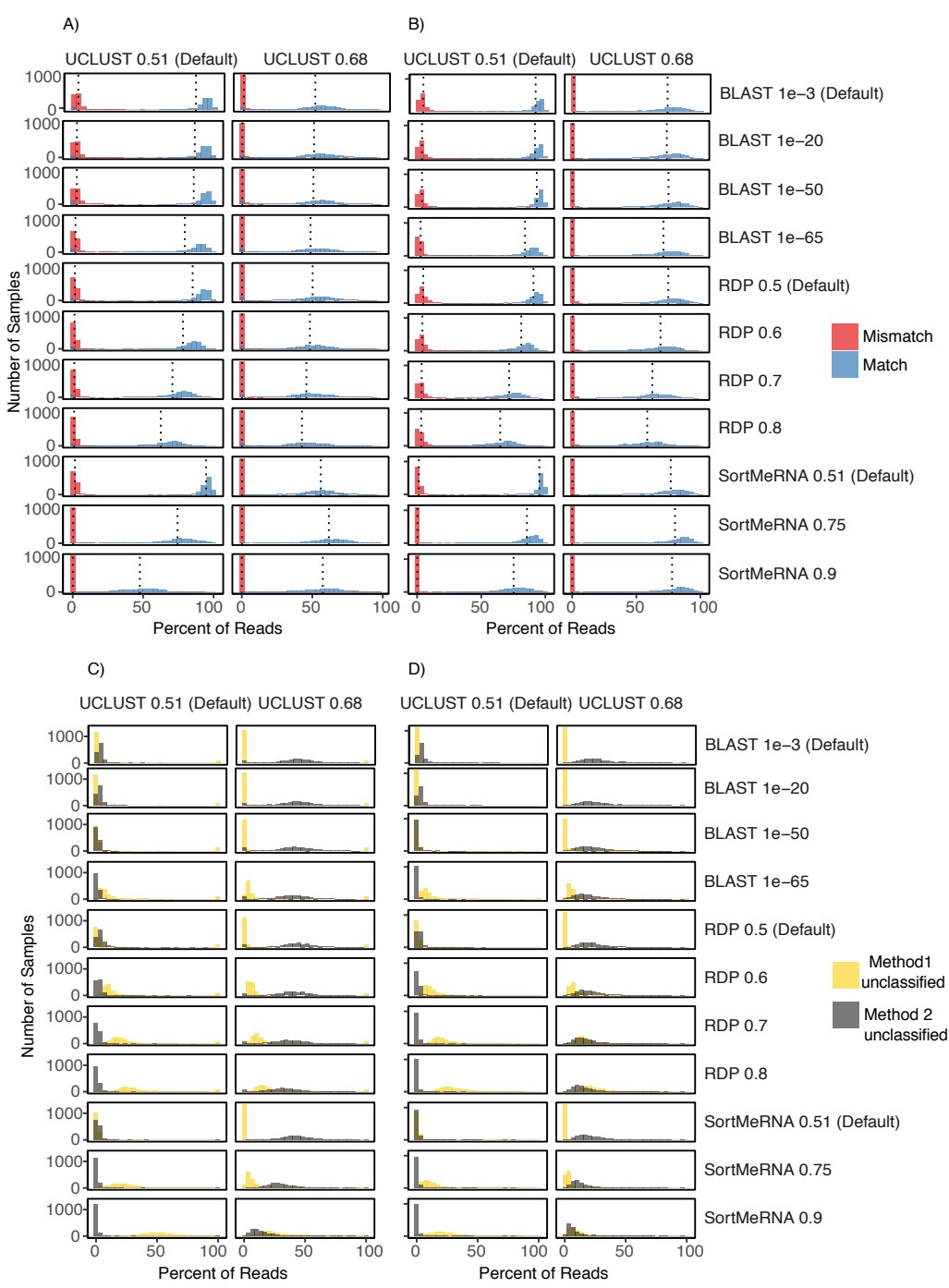

**Figure 4** **Comparison of UCLUST classification results to other classifier results across the different parameter values and using both the Greengenes (A) and (C) and HITdb (B) and (D) databases.** Histograms are shown for the comparison metrics between samples at the match and mismatch categories (A) and (B) and for the unclassified by one method but not the other categories (C) and (D). In (C) and (D) Method 1 unclassified are row labels and Method 2 unclassified are (continued on next page...)

**Figure 4 (…continued)**
column labels. For illustrative purposes, the UCLUST Default and SortMeRNA 0.9 comparison between unclassified reads and using the Greengenes database (D) resulted in virtually no reads assigned to a taxon with UCLUST that were not also assigned with SortMeRNA (Method2Nulls) but some reads were assigned to a taxon with UCLUST but not with SortMeRNA (which is to be expected as SortMeRNA 0.9 is very stringent). See Supplemental Information 1 for all comparisons.

## Impact of varying confidence threshold and reference database

The level of congruence among the classifiers varied substantially depending on the parameter settings that each used to determine whether a read was classified to a taxon. However, no parameter combinations produced complete congruence between two classifiers and, in general, the default settings resulted in the highest congruence among the classifiers (fewest number of reads classified by one method but not the other and highest number of reads being classified to the same taxon (i.e., match; Figs. 2 and 4; Supplemental Information 1). An example of classification at the family rank with the RDP classifier and the Greengenes database illustrates this. As the cutoff for the confidence parameter for RDP is increased from the default, a greater number of reads were not classified; thus, fewer reads were classified to the same taxon (Figs. 4A & 4B). This suggests that a higher threshold setting within RDP has the unwanted consequence of producing many false negative assignments (a read is not classified to a taxon when it should be).

The pattern that different threshold parameter cutoffs affected the degree of congruence among the assigners and that no combination of threshold settings resulted in complete congruence was also observed when classifying reads based on the HITdb database (Fig. 4; Supplemental Information 1).

## CONCLUSIONS

Citizen science projects, like the American Gut Project, represent a great opportunity to further public health. We had initially thought to use the publicly available data, specifically the assignment of reads to known foodborne pathogens and the rich metadata about each sample, to generate hypotheses about the prevalence of such pathogens within the general population and examine potential lifestyle and demographic correlates with carrier status. The results of taxonomic assignment to each of the reads within the AGP showed this to be a promising avenue to pursue, as ∼15% of the samples were 'positive' for *Listeria* or *Salmonella*. Before making claims about these pathogens that would be relevant to public health we sought to evaluate the robustness of those classifications by re-analyzing the positive data and evaluating the taxonomy assignments using additional classifiers. Although we do not know the true taxonomy of the reads being classified, the results of those additional analyses that showed many reads being classified differently by the classifiers raise questions about how reliable those classifications are. Those results also caused us to forego any analyses looking into correlates in the metadata that may explain the presence of *Listeria* or *Salmonella* within an individual's skin, mouth, and/or fecal microbiome. The incongruence we observed among the methods (i.e., Segal's Law) is due, in part, to the different underlying assumptions and algorithms employed in each classifier.

It is also partly due to the V4 16S rRNA marker potentially lacking the information content (especially at lower taxonomic ranks like species) to be robust to those differences in algorithms and assumptions. Interestingly, the disagreement between methods cannot be removed or made insignificant by adjustments to the confidence threshold or by using a database designed for human enterobacteria; we explored this possibility, and there were always an appreciable number of reads with either mismatched taxon classification or no classification for one of the methods (Fig. 4).

Our results are in agreement with those published elsewhere on the negative relationship between accuracy in classifying V4 reads and decreasing taxonomic rank (*Clarridge 3rd, 2004*; *Liu et al., 2008*; *Srinivasan et al., 2015*). However, the V4 region has been shown to best capture the information content contained in the full 16S rRNA gene and to have the best sensitivity for bacterial and phylogenetic analyses (*Wang et al., 2007*; *Yang, Wang & Qian, 2016*). Our results are reassuring in that read classification matches greatly outnumber mismatches (Figs. 2 and 4). However, there are a large number of instances in which a read is classified by one method but not another (Figs. 2 and 4); at the family level, 5% of reads were classified by one method but not the other and this increases to 16% and 40% at the genus and species levels, respectively. These problems make it difficult to use this data for research in *Salmonella* and *Listeria*, and certainly detract from its use in a regulatory setting. Additionally, the results presented here showing that even at the family level there can be 5% of reads classified by one method and not the other show that conclusions within 16S rRNA gene studies and, likely, metagenome-wide association studies (*Wang & Jia, 2016*) health policy should be set based on good empirical information, and misleading pathogen detection is problematic. This highlights the difficulty of associating pathogens with a given phenotype (e.g., obesity; *Sze & Schloss, 2016*) with the 16S rRNA gene V4 region using current taxonomic assigners, and points toward problems that have been remarked upon elsewhere regarding the publication of results obtained with a method that is not fit for purpose (e.g., *Gonzalez et al., 2016*). There is a clear need for improved methods that provide measures of taxonomic assignment uncertainty and can support good public health policy decision making.

## ACKNOWLEDGEMENTS

We thank A Shpuntoff for assistance with conducting the analyses on the high performance computing cluster and FDA Center for Food Safety and Applied Nutrition for supporting this research. We also thank M Ferguson for insightful discussion on methods for assessing inter-rater agreement.

### Funding
The authors received no funding for this work.

### Competing Interests
The authors declare there are no competing interests.

## Author Contributions

- James B. Pettengill conceived and designed the experiments, performed the experiments, analyzed the data, contributed reagents/materials/analysis tools, wrote the paper, prepared figures and/or tables, reviewed drafts of the paper.
- Hugh Rand conceived and designed the experiments, wrote the paper, reviewed drafts of the paper.

## Data Availability

The raw data and code used to classify the reads are both publicly available at the AGP and QIIME websites, respectively, as noted in the text.

## Supplemental Information

Supplemental information for this article can be found online at http://dx.doi.org/10.7717/peerj.3480#supplemental-information.

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
