# Peer review of "Segal’s Law, 16S rRNA gene sequencing, and the perils of foodborne pathogen detection within the American Gut Project"

_PeerJ, doi:10.7717/peerj.3480_

## Round 0.1 · original submission · Major Revisions

· Academic Editor

Major Revisions

As noted by the reviewers there are a number of issues that need to be addressed. Some of these are grammatical and some will make the manuscript more readable, such as the suggestion to define Segal's law upfront.

The principle reasons for needing a major revision are the need to address the experimental decision as picked up by both reviewers, some of which may simply be due to a lack of detail in the current methods section. The question of different outcomes using different databases is also an important one, and some reasoning at least over the choice of database should be given.

·

Basic reporting

"Segal's Law, amplicon metagenomics and the perils of foodborne pathogen detection within the American Gut Project" is a very well-written and easy to read manuscript. I think it also addresses an important problem and has the potential to be very instructive.

While this is preference, I would suggest that the authors define Segal's Law in the abstract and introduction more directly, rather than just referencing it in parentheses. Since it is part of the title, the definition should be emphasized a bit more by making it the subject of the sentence.

Citations and figures are reasonable.

The study is self-contained.

Minor points:
*The term "16S rRNA gene sequencing" or similar should be used instead of just "16S" in several spots.
*The "s" in "16S" should be capitalized.
*16S rRNA gene sequencing is not metagenomics, so "metagenomics" and "metagenome-wide association" should be removed from the title and the text when referring to 16S rRNA gene analysis.
*In the abstract, clarify the comment that the V4 region does not provide good resolution at the lower taxonomic levels. Its resolution can be quite good, but there are limitations to any 16S rRNA gene amplicon study. Are you referring to specific organisms? Species-level?
*In the introduction, it is not clear what the numbers in parentheses are in the sentence reporting incidence of disease.
*Intro line 67 - "or worse" - it would be better to be more specific
*SortMeRNA should not be all caps (authors named it with some lowercase)

Experimental design

The article is within the scope of the journal.

The research question is very well-defined, and they address a known limitation of 16S rRNA gene sequencing.

There are some limitations in the methods used to compare the classification. More detail should be given to the methods, and some of the differences in the algorithms need to be taken into account when interpreting the results.
*default parameters for each program need to be defined.
*reporting of the alignments is not necessarily comparable.
--For instance, RDP will assign a confidence value for classification at each taxonomic level. Did you only use those with high confidence (0.8 is a common cut off)? How many iterations were done in your default method?
--RDP will also classify things to a higher taxonomic level at high confidence even when it fails to classify to a lower taxonomic level. BLAST, though, does not have a method to do that. It can report alignment with a high degree of mismatch -- so even if its best hit is not an exact (or even especially close) match, you may get a report that says they align. Are you interpreting that as "classified"?
--Along these lines, the differences in each of the methods and their reporting need to be described in more detail.
*The differences in classification of Listeria and Salmonella by each method should be picked apart a little more so that we understand why there are differences. For instance, with RDP, because classification is assigned a confidence value and cut offs are typically applied, you can have a sequence that is near the cut off that can land on "classified genus" in one run and "classified family" in another just because of the results of that set of bootstrap iterations. Does any of the disagreement come from sequences classified at a confidence that just misses the cut off threshold? Is BLAST over classifying and assigning sequences with quite a few mismatches? Results of others?

Validity of the findings

There is no question that the different methods of classification will yield different results, so the authors' general findings are reasonable. The methods need to be clarified a little before we can fully interpret the results presented here.

I may be misunderstanding something about figure 2, but it seems unusual to me that there are so many things not classified by BLAST (not classified Method 2) compared to RDP since BLAST seemed to perform more robustly at the species level for the Listeria and Salmonella and in Figure 1. Conceptually, I would expect that to to be the case for other organisms as well.

There is often not enough information in the V4 region to classify to that taxonomic level. Confidence at species level would be limited to some taxa and dependent on having appropriate references in the reference database. This should be noted as a general, known limitation.

Additional comments

The idea for using the American Gut Project data to look for prevalence of food-borne pathogens is a good one. The study makes an important point: different results can certainly be obtained from different analysis approaches. While this is not unknown or unaddressed by data analysts, the authors describe a very interesting and useful case. The methods need to be better described and considered, and this may rapidly lead to a better understanding of the problem that ultimately allows them to determine whether the sequences they are interested in are represented in the data sets.

There are reasons why the outputs of the algorithms differ, and it is likely that the results are different in this study due to use of default parameters and (possibly) the need to parse output files to filter the high-confidence results. In addition, there are known limitations associated with classifying to species level. For food-borne pathogens, the authors would want to determine that (1) there were sufficient species represented in the reference db for classification and (2) that the species had sequences that were distinct. If true, then they likely can identify and classify sequences, but it will take some manual review in order to be confident in the results.

Reviewer 2 ·

Basic reporting

The presented analyses sounds with three high quality figures included. The manuscript is well written; the authors have used standard Scientific English throughout the text and properly cited references when needed.

Experimental design

I am a bit concerned about few issues, and that include the study design. Please see my comments below:

1. The authors have used Greengenes database, the one used in AGP, as a reference database. But shall we expect different outcomes by using other databases. For instance, RDP [PMC3965039], SILVA [PMC3531112] or the annotation database contained in MG-RAST [PMC2563014]).

2. The authors should also consider the usage of more specialized databases that comprising members of human gut microbiomes (e.g. PMC4676846 and PMC4702862). The use of more specialized database has been found to increase the resolution of taxonomic classification and therefore would provide further evidences to support the conclusion drawn form the current analysis.

3. Line 101, the authors have used the default parameters with each classifier they have used in their analysis; I would check the similarity threshold used by each classifier. Similarity threshold of 99% is generally acceptable but even this high threshold is often insufficient to discriminate between closely related members of Enterobacteriaceae.

4. The authors should comment on the quality of the data used in the analysis. What was the quality threshold used? Have they excluded sequences because of quality requirement? And whether different primers have been used to amplify the V4 region?

Validity of the findings

no comment.

Additional comments

The authors have used selected taxonomy classifiers to identify the abundance of Salmonella and Listeria reads in AGP samples. They have concluded that the taxonomic assignment is greatly varied depending on the algorithm used in analyses.

1. The objective of the study is not that clear to me. I believe that the main objective of the study was to evaluate the impact of using different algorithms (read classifiers) on the taxonomic assignment of 16S-based metagenomic data. If this was the case; then lines, 65-76 require rephrasing.

2. Lines 23 and 24, the authors mentioned about the variations identified using different read classifiers without providing any further information. This information should be summarized in Abstract, and details should be given in appropriate sections of the manuscript.

3. Lines 52-54, citizen science initiative, particularly those use metagenomics, cannot be an alternative approach to symptomatic surveillance. Theses initiatives might help in identifying correlations between human micrbiome and health status or provide approximate estimates of microbiom structure (signatures) in particular a community/group. I would rephrase this paragraph to emphasize more on the challenges associated with this approach (e.g. the bioinformatics challenges and data misinterpretation and validation issues).

---

## Round 0.2 · Minor Revisions

· Academic Editor

Minor Revisions

Please see comments from Reviewer 1. In particular it is requested that you look at their requirement for some concrete explanations for while the differences observed may be occurring.

·

Basic reporting

OK

Experimental design

The authors improved the description of parameters and databases used. As I said before, they describe a well-known issue with sequence classification: classification methods differ.

What I was trying to get at with my previous comments was that the paper would be greatly strengthened by offering a few concrete explanations for why some of these differences occur. I don't think the authors have addressed that. I was not proposing using fuzzy cutoffs for analysis, but rather trying to understand whether the selected cutoffs can be demonstrated to be one cause for discrepancy between methods.

For instance, specifically looking at the Listeria and Salmonella reads that are classified differently by each method could shed light on this. It does not mean that the analysis needs to be run multiple times; it means that for reads that are classified differently, the confidence score for the RDP classifier or the e-value for BLAST, etc. would need to be examined. For BLAST in particular, the e-value may not be the best cutoff because it varies with the size of the database and factors other than the quality of your alignment. Maybe you need to use a bitscore cutoff appropriate for your read length, and that might become clear if you evaluate specific reads.

Regarding my comment about the differences in classification between RDP and BLAST, you should add more detail about how each method reported/parsed taxonomic results. To clarify my previous comment, RDP will classify some reads to, for example, the Phylum level with 0.5 confidence but not be able to classify the read further with high confidence. Do you count those reads or only fully classified reads? For BLAST with an e-value cutoff, how do you assign taxonomy? I think people frequently take the full taxonomy from whatever reference was hit. So do you only have taxonomy for fully classified reads? Or can you have partially classified reads? Please clarify in methods. These kinds of differences can offer valuable insights into discrepancies between classification methods.

Validity of the findings

Good. Please re-emphasize in discussion that there was no "true positive" for Listeria and Salmonella used in the testing, so this is just a comparison of methods and not an assessment of the ability of the methods to detect a known pathogen.

Reviewer 2 ·

Basic reporting

no comment

Experimental design

no comment

Validity of the findings

no comment

Additional comments

no comment

---

## Round 0.3 · accepted · Accept

· Academic Editor

Accept

Thank you for addressing the minor concerns of the reviewer. I note that you have addressed all of the outstanding concerns.